# Dynamic Regret of Convex and Smooth Functions

**Peng Zhao, Yu-Jie Zhang, Lijun Zhang, Zhi-Hua Zhou**
National Key Laboratory for Novel Software Technology,
Nanjing University, Nanjing 210023, China
{zhaop, zhangyj, zhanglj, zhouzh}@lamda.nju.edu.cn

## Abstract

We investigate online convex optimization in non-stationary environments and choose the *dynamic regret* as the performance measure, defined as the difference between cumulative loss incurred by the online algorithm and that of any feasible comparator sequence. Let $T$ be the time horizon and $P_T$ be the path-length that essentially reflects the non-stationarity of environments, the state-of-the-art dynamic regret is $\mathcal{O}(\sqrt{T(1 + P_T)})$. Although this bound is proved to be minimax optimal for convex functions, in this paper, we demonstrate that it is possible to further enhance the dynamic regret by exploiting the smoothness condition. Specifically, we propose novel online algorithms that are capable of leveraging smoothness and replace the dependence on $T$ in the dynamic regret by *problem-dependent* quantities: the variation in gradients of loss functions, the cumulative loss of the comparator sequence, and the minimum of the previous two terms. These quantities are at most $\mathcal{O}(T)$ while could be much smaller in benign environments. Therefore, our results are adaptive to the intrinsic difficulty of the problem, since the bounds are tighter than existing results for easy problems and meanwhile guarantee the same rate in the worst case.

## 1 Introduction

In many real-world applications, data are inherently accumulated over time, and thus it is of great importance to develop a learning system that updates in an online fashion. Online Convex Optimization (OCO) is a powerful paradigm for learning in such a circumstance, which can be regarded as an iterative game between a player and an adversary. At iteration $t$, the player selects a decision $\mathbf{x}_t$ from a convex set $\mathcal{X}$ and the adversary reveals a convex function $f_t : \mathcal{X} \mapsto \mathbb{R}$. The player subsequently suffers an instantaneous loss $f_t(\mathbf{x}_t)$. The performance measure is the (static) *regret* [1],

$$\text{S-Regret}_T = \sum_{t=1}^{T} f_t(\mathbf{x}_t) - \min_{\mathbf{x} \in \mathcal{X}} \sum_{t=1}^{T} f_t(\mathbf{x}), \tag{1}$$

which is the difference between cumulative loss incurred by the online algorithm and that of the best decision in hindsight. The rationale behind such a metric is that the best fixed decision in hindsight is reasonably good over all the iterations. However, this is too optimistic and may not hold in changing environments, where data are evolving and the optimal decision is drifting over time. To address this limitation, *dynamic regret* is proposed to compete with changing comparators $\mathbf{u}_1, \ldots, \mathbf{u}_T \in \mathcal{X}$,

$$\text{D-Regret}_T(\mathbf{u}_1, \ldots, \mathbf{u}_T) = \sum_{t=1}^{T} f_t(\mathbf{x}_t) - \sum_{t=1}^{T} f_t(\mathbf{u}_t), \tag{2}$$

which draws considerable attention recently [2, 3, 4, 5, 6, 7, 8, 9, 10, 11, 12, 13]. The measure is also called the *universal* dynamic regret (or *general* dynamic regret), in the sense that it gives a universal

guarantee that holds against *any* comparator sequence. Note that static regret (1) can be viewed as its special form by setting comparators as the fixed best decision in hindsight. Moreover, a variant appeared frequently in the literature is the *worst-case* dynamic regret defined as

$$\text{D-Regret}_T(\mathbf{x}_1^*, \dots, \mathbf{x}_T^*) = \sum_{t=1}^{T} f_t(\mathbf{x}_t) - \sum_{t=1}^{T} f_t(\mathbf{x}_t^*), \tag{3}$$

which specializes the general form (2) by setting $\mathbf{u}_t = \mathbf{x}_t^* \in \arg\min_{\mathbf{x} \in \mathcal{X}} f_t(\mathbf{x})$. However, the worst-case dynamic regret is often too pessimistic, whereas the universal one is more adaptive to the non-stationary environments. We refer the readers to [8] for more detailed explanations.

There are many studies on the worst-case dynamic regret [2, 3, 4, 5, 6, 7, 10, 13], but only few results are known for the universal dynamic regret. Zinkevich [1] shows that online gradient descent (OGD) achieves an $\mathcal{O}(\sqrt{T}(1 + P_T))$ universal dynamic regret, where $P_T = \sum_{t=2}^{T} \|\mathbf{u}_{t-1} - \mathbf{u}_t\|_2$ is the path-length of comparators $\mathbf{u}_1, \dots, \mathbf{u}_T$ and thus reflects the non-stationarity of the environments. Nevertheless, there exists a large gap between this upper bound and the $\Omega(\sqrt{T(1 + P_T)})$ minimax lower bound established recently by Zhang et al. [8], who further propose a novel online algorithm, attaining an $\mathcal{O}(\sqrt{T(1 + P_T)})$ universal dynamic regret, and thereby close the gap.

Although the rate is minimax optimal for convex functions, we would like to design algorithms with more adaptive bounds, replacing the dependence on $T$ by certain *problem-dependent* quantities that are $\mathcal{O}(T)$ in the worst case while could be much smaller in benign environments (i.e., easy problems). In the study of static regret, we can attain such bounds when additional curvature like smoothness is presented, including small-loss bounds [14] and variation bounds [15]. Thus, a natural question arises *whether it is possible to leverage smoothness to achieve more adaptive universal dynamic regret?*

**Our results.** In this paper, we provide an affirmative answer by designing algorithms with problem-dependent dynamic regret bounds. Specifically, we focus on the following two adaptive quantities: the gradient variation of online functions $V_T$ and the cumulative loss of the comparator sequence $F_T$,

$$V_T = \sum_{t=2}^{T} \sup_{\mathbf{x} \in \mathcal{X}} \|\nabla f_{t-1}(\mathbf{x}) - \nabla f_t(\mathbf{x})\|_2^2, \text{ and } F_T = \sum_{t=1}^{T} f_t(\mathbf{u}_t). \tag{4}$$

We propose a novel online approach for convex and smooth functions, named Smoothness-aware online learning with dynamic regret (abbreviated as Sword). There are three versions, including $\text{Sword}_{\text{var}}$, $\text{Sword}_{\text{small}}$, and $\text{Sword}_{\text{best}}$. All of them enjoy problem-dependent dynamic regret bound:

- $\text{Sword}_{\text{var}}$ enjoys a gradient-variation bound of $\mathcal{O}(\sqrt{(1 + P_T + V_T)(1 + P_T)})$;
- $\text{Sword}_{\text{small}}$ enjoys a small-loss bound of $\mathcal{O}(\sqrt{(1 + P_T + F_T)(1 + P_T)})$;
- $\text{Sword}_{\text{best}}$ enjoys a best-of-both-worlds bound of $\mathcal{O}(\sqrt{(1 + P_T + \min\{V_T, F_T\})(1 + P_T)})$.

Comparing to the minimax rate of $\mathcal{O}(\sqrt{T(1 + P_T)})$, our bounds replace the dependence on $T$ by the problem-dependent quantity $P_T + \min\{V_T, F_T\}$. Since the quantity is at most $\mathcal{O}(T)$, our bounds become much tighter when the problem is easy (for example when $P_T$ and $V_T/F_T$ are sublinear in $T$), and meanwhile safeguard the same guarantee in the worst case. Therefore, our results are adaptive to the intrinsic difficulty of the problem and the non-stationarity of the environments.

**Technical contributions.** We highlight challenges and technical contributions of this paper. First, we note that there exist studies showing that the worst-case dynamic regret can benefit from smoothness [5, 6, 13]. However, their analyses do not apply to our case, since we cannot exploit the *optimality condition* of comparators $\mathbf{u}_1, \dots, \mathbf{u}_T$, in stark contrast with the worst-case dynamic regret analysis. Therefore, we adopt the meta-expert framework to *hedge the non-stationarity while keeping the adaptivity*. We can use variants of OGD as the expert-algorithm to exploit the smoothness, but it is difficult to design an appropriate meta-algorithm. Existing meta-algorithms and their variants either lead to problem-independent regret bounds or introduce terms that are incompatible to the desired problem-dependent quantity. To address the difficulty, we adopt the technique of *optimistic online learning* [16, 17], in particular OptimisticHedge, to design novel meta-algorithms.

For $\text{Sword}_{\text{var}}$, we apply OptimisticHedge with carefully designed optimism, which allows us to exploit the negative term in the regret analysis of OptimisticHedge [17]. In this way, the meta-regret only

depends on the gradient variation. The construction of the special optimism is the most challenging part of our paper. For Sword$_{\text{small}}$, the design of meta-algorithm is simple, and we directly use the vanilla Hedge, which can be treated as OptimisticHedge with null optimism. Finally, for Sword$_{\text{best}}$, we still employ OptimisticHedge as the meta-algorithm, but introduce a parallel meta-algorithm to *learn the best optimism* to ensure a best-of-both-worlds dynamic regret guarantee.

## 2 Gradient-Variation and Small-Loss Bounds

We first list assumptions used in the paper, and then propose online algorithms with gradient-variation and small-loss dynamic regret bounds, respectively. All the proofs can be found in the full paper [18].

### 2.1 Assumptions

We introduce the following common assumptions that might be used in the theorems.

**Assumption 1.** The norm of the gradients of online functions over the domain $\mathcal{X}$ is bounded by $G$, i.e., $\|\nabla f_t(\mathbf{x})\|_2 \leq G$, for all $\mathbf{x} \in \mathcal{X}$ and $t \in [T]$.

**Assumption 2.** The domain $\mathcal{X} \subseteq \mathbb{R}^d$ contains the origin $\mathbf{0}$, and the diameter of the domain $\mathcal{X}$ is at most $D$, i.e., $\|\mathbf{x} - \mathbf{x}'\|_2 \leq D$ for any $\mathbf{x}, \mathbf{x}' \in \mathcal{X}$.

**Assumption 3.** All the online functions are $L$-smooth, i.e., for any $\mathbf{x}, \mathbf{x}' \in \mathcal{X}$ and $t \in [T]$,

$$\|\nabla f_t(\mathbf{x}) - \nabla f_t(\mathbf{x}')\|_2 \leq L\|\mathbf{x} - \mathbf{x}'\|_2. \tag{5}$$

**Assumption 4.** All the online functions are non-negative.

Note that in Assumption 4 we require the online functions to be non-negative outside the domain $\mathcal{X}$, which is a precondition for establishing the self-bounding property for smooth functions [14]. Meanwhile, we treat double logarithmic factors in $T$ as a constant, following previous studies [19, 20].

### 2.2 Gradient-Variation Bound

We design an approach in a meta-expert framework, and prove its gradient-variation dynamic regret.

#### 2.2.1 Expert-Algorithm

In the study of static regret, Chiang et al. [15] propose the following online extra-gradient descent (OEGD) algorithm, and show that the algorithm enjoys gradient-variation static regret bound. The OEGD algorithm performs the following update:

$$\begin{aligned}
\widehat{\mathbf{x}}_{t+1} &= \Pi_{\mathcal{X}}\left[\widehat{\mathbf{x}}_t - \eta \nabla f_t(\mathbf{x}_t)\right], \\
\mathbf{x}_{t+1} &= \Pi_{\mathcal{X}}\left[\widehat{\mathbf{x}}_{t+1} - \eta \nabla f_t(\widehat{\mathbf{x}}_{t+1})\right],
\end{aligned} \tag{6}$$

where $\widehat{\mathbf{x}}_1, \mathbf{x}_1 \in \mathcal{X}$, $\eta > 0$ is the step size, and $\Pi_{\mathcal{X}}[\cdot]$ denotes the projection onto the nearest point in $\mathcal{X}$. For convex and smooth functions, Chiang et al. [15] prove that OEGD achieves an $\mathcal{O}(\sqrt{V_T})$ static regret. We further demonstrate that OEGD also enjoys gradient-variation type dynamic regret.

**Theorem 1.** *Under Assumptions 1, 2, and 3, by choosing $\eta \leq \frac{1}{4L}$, OEGD (6) satisfies*

$$\sum_{t=1}^{T} f_t(\mathbf{x}_t) - \sum_{t=1}^{T} f_t(\mathbf{u}_t) \leq \frac{D^2 + 2DP_T}{2\eta} + \eta V_T + GD = \mathcal{O}\left(\frac{1 + P_T}{\eta} + \eta V_T\right).$$

*for* any *comparator sequence* $\mathbf{u}_1, \ldots, \mathbf{u}_T \in \mathcal{X}$.

Theorem 1 shows that it is crucial to tune the step size to balance non-stationarity (path-length $P_T$) and adaptivity (gradient variation $V_T$). Notice that the optimal tuning $\eta^* = \sqrt{(D^2 + 2DP_T)/(2V_T)}$ requires the prior information of $P_T$ and $V_T$ that are generally unavailable. We emphasize that $V_T$ is empirically computable, while $P_T$ remains unknown even after all iterations due to the fact that the comparator sequence is unknown and can be chosen arbitrarily as long as it is feasible in the domain. Therefore, the doubling trick [21] can only remove the dependence on the unknown $V_T$ but not $P_T$.

To handle the uncertainty, we adopt the meta-expert framework to *hedge the non-stationarity while keeping the adaptivity*, inspired by the recent advance in learning with multiple learning rates [22,

23, 8]. Concretely, we first construct a pool of candidate step sizes to discretize value range of the optimal step size, and then initialize multiple experts simultaneously, denoted by $\mathcal{E}_1, \ldots, \mathcal{E}_N$. Each expert $\mathcal{E}_i$ returns its prediction $\mathbf{x}_{t,i}$ by running OEGD (6) with a step size $\eta_i$ from the pool. Finally, predictions of all the experts are combined by a meta-algorithm as the final output $\mathbf{x}_t$ to track the best expert. From the procedure, we observe that the dynamic regret can be decomposed as,

$$\text{D-Regret}_T = \sum_{t=1}^{T} f_t(\mathbf{x}_t) - \sum_{t=1}^{T} f_t(\mathbf{u}_t) = \underbrace{\sum_{t=1}^{T} f_t(\mathbf{x}_t) - f_t(\mathbf{x}_{t,i})}_{\texttt{meta-regret}} + \underbrace{\sum_{t=1}^{T} f_t(\mathbf{x}_{t,i}) - f_t(\mathbf{u}_t)}_{\texttt{expert-regret}},$$

where $\{\mathbf{x}_t\}_{t=1,\ldots,T}$ denotes the final output sequence, and $\{\mathbf{x}_{t,i}\}_{t=1,\ldots,T}$ is the prediction sequence of expert $\mathcal{E}_i$. The first part is the difference between cumulative loss of final output sequence and that of prediction sequence of expert $\mathcal{E}_i$, which is introduced by the meta-algorithm and thus named as *meta-regret*; the second part is the dynamic regret of expert $\mathcal{E}_i$ and therefore named as *expert-regret*.

The expert-algorithm is set as OEGD (6), and Theorem 1 upper bounds the expert-regret. The main difficulty lies in the design and analysis of an appropriate meta-algorithm.

### 2.2.2 Meta-Algorithm

Formally, there are $N$ experts and expert $\mathcal{E}_i$ predicts $\mathbf{x}_{t,i}$ at iteration $t$, and the meta-algorithm requires to produce $\mathbf{x}_t = \sum_{i=1}^{N} p_{t,i}\mathbf{x}_{t,i}$, a weighted combination of expert predictions, where $\boldsymbol{p}_t \in \Delta_N$ is the weight vector. It is natural to use Hedge [24] for weight update in order to track the best expert.

In order to be compatible to the gradient-variation expert-regret, the meta-algorithm is required to incur a problem-dependent meta-regret of order $\mathcal{O}(\sqrt{V_T \ln N})$. However, the meta-algorithms used in existing studies [23, 8] cannot satisfy the requirements. For example, the vanilla Hedge (multiplicative weights update) suffers from an $\mathcal{O}(\sqrt{T \ln N})$ meta-regret, which is problem-independent and thus not suitable for us. To this end, we design a a novel variant of Hedge by leveraging the technique of *optimistic online learning* with carefully designed optimism, specifically for our problem.

The optimistic online learning is developed by Rakhlin and Sridharan [16] and further expanded by Syrgkanis et al. [17]. For the prediction with expert advice setting, they consider that at the beginning of iteration $(t + 1)$, in addition to the loss vector $\boldsymbol{\ell}_t \in \mathbb{R}^N$ returned by the experts, the learner can receive a vector $\boldsymbol{m}_{t+1} \in \mathbb{R}^N$ called *optimism*. The authors propose the OptimisticHedge algorithm [16, 17], which updates the weight vector $\boldsymbol{p}_{t+1} \in \Delta_N$ by

$$p_{t+1,i} \propto \exp\left(-\varepsilon\Big(\sum_{s=1}^{t} \ell_{s,i} + m_{t+1,i}\Big)\right), \quad \forall i \in [N]. \tag{7}$$

Syrgkanis et al. [17] prove the following regret guarantee for OptimisticHedge.

**Lemma 1** ([17, Theorem 19]). *The meta-regret of OptimisticHedge is upper bounded by*

$$\sum_{t=1}^{T} \langle \boldsymbol{p}_t, \boldsymbol{\ell}_t \rangle - \ell_{t,i} \leq \frac{2 + \ln N}{\varepsilon} + \varepsilon \sum_{t=1}^{T} \|\boldsymbol{\ell}_t - \boldsymbol{m}_t\|_{\infty}^2 - \frac{1}{4\varepsilon} \sum_{t=2}^{T} \|\boldsymbol{p}_t - \boldsymbol{p}_{t-1}\|_1^2, \tag{8}$$

*which holds for any expert $i \in [N]$. Denote by $D_{\infty} = \sum_{t=1}^{T} \|\boldsymbol{\ell}_t - \boldsymbol{m}_t\|_{\infty}^2$ to measure the adaptivity. With proper learning rate tuning, OptimisticHedge enjoys an $\mathcal{O}(\sqrt{D_{\infty} \ln N})$ meta-regret.*

The optimistic online learning is very powerful for designing adaptive methods, in that the adaptivity $D_{\infty}$ in Lemma 1 is very general and can be specialized flexibly with different configurations of the feedback loss $\boldsymbol{\ell}_t$ and optimism $\boldsymbol{m}_t$. Based on the OptimisticHedge, we propose VariationHedge, the meta-algorithm for Sword$_{\text{var}}$, by specializing OptimisticHedge as follows:

- the feedback loss $\boldsymbol{\ell}_t$ is set as the linearized surrogate loss, namely, $\ell_{t,i} = \langle \nabla f_t(\mathbf{x}_t), \mathbf{x}_{t,i} \rangle$;
- the optimism $\boldsymbol{m}_t$ is set with a careful design: for each $i \in [N]$

$$m_{t,i} = \langle \nabla f_{t-1}(\bar{\mathbf{x}}_t), \mathbf{x}_{t,i} \rangle, \text{ where } \bar{\mathbf{x}}_t = \sum_{i=1}^{N} p_{t-1,i}\mathbf{x}_{t,i}. \tag{9}$$

| **Algorithm 1** Sword$_{\text{var}}$: Meta (VariationHedge) | **Algorithm 2** Sword$_{\text{var}}$: Expert (OEGD) |
|---|---|
| **Input:** step size pool $\mathcal{H}_{\text{var}}$; learning rate $\varepsilon$ | **Input:** step size $\eta_i$ |
| 1: Initialization: $\forall i \in [N], p_{0,i} = 1/N$ | 1: Let $\widehat{\mathbf{x}}_{1,i}, \mathbf{x}_{1,i}$ be any point in $\mathcal{X}$ |
| 2: **for** $t = 1$ **to** $T$ **do** | 2: **for** $t = 1$ **to** $T$ **do** |
| 3:   Receive $\mathbf{x}_{t+1,i}$ from expert $\mathcal{E}_i\,(\eta_i)$ | 3:   $\widehat{\mathbf{x}}_{t+1,i} = \Pi_{\mathcal{X}}\big[\widehat{\mathbf{x}}_{t,i} - \eta_i \nabla f_t(\mathbf{x}_{t,i})\big]$ |
| 4:   Update weight $p_{t+1,i}$ by (10) | 4:   $\mathbf{x}_{t+1,i} = \Pi_{\mathcal{X}}\big[\widehat{\mathbf{x}}_{t+1,i} - \eta_i \nabla f_t(\widehat{\mathbf{x}}_{t+1,i})\big]$ |
| 5:   Predict $\mathbf{x}_{t+1} = \sum_{i=1}^{N} p_{t+1,i}\mathbf{x}_{t+1,i}$ | 5:   Send $\mathbf{x}_{t+1,i}$ to meta-algorithm |
| 6: **end for** | 6: **end for** |

So the meta-algorithm of Sword$_{\text{var}}$ (namely, VariationHedge) updates the weight by

$$p_{t+1,i} \propto \exp\left(-\varepsilon\Big(\sum_{s=1}^{t}\langle \nabla f_s(\mathbf{x}_s), \mathbf{x}_{s,i}\rangle + \langle \nabla f_t(\bar{\mathbf{x}}_{t+1}), \mathbf{x}_{t+1,i}\rangle\Big)\right), \quad \forall i \in [N]. \qquad (10)$$

Algorithm 1 summarizes detailed procedures of the meta-algorithm, which in conjunction with the expert-algorithm of Algorithm 2 yields the Sword$_{\text{var}}$ algorithm.

**Remark 1.** The design of optimism in (9) (in particular, $\bar{\mathbf{x}}_t$) is crucial, and is the most challenging part in this work. The key idea is to exploit the negative term in the regret of OptimisticHedge, as shown in (8), to convert the adaptive quantity $D_\infty$ to the desired gradient variation $V_T$. Indeed,

$$\|\boldsymbol{\ell}_t - \boldsymbol{m}_t\|_\infty^2 \overset{(9)}{=} \max_{i\in[N]}\langle \nabla f_t(\mathbf{x}_t) - \nabla f_{t-1}(\bar{\mathbf{x}}_t), \mathbf{x}_{t,i}\rangle^2$$
$$\leq D^2\|\nabla f_t(\mathbf{x}_t) - \nabla f_{t-1}(\bar{\mathbf{x}}_t)\|_2^2$$
$$\leq 2D^2(\|\nabla f_t(\mathbf{x}_t) - \nabla f_{t-1}(\mathbf{x}_t)\|_2^2 + \|\nabla f_{t-1}(\mathbf{x}_t) - \nabla f_{t-1}(\bar{\mathbf{x}}_t)\|_2^2)$$
$$\leq 2D^2 \sup_{\mathbf{x}\in\mathcal{X}}\|\nabla f_t(\mathbf{x}) - \nabla f_{t-1}(\mathbf{x})\|_2^2 + 2D^2 L^2 \|\mathbf{x}_t - \bar{\mathbf{x}}_t\|_2^2$$

where the last step makes use of smoothness. Therefore, $D_\infty$ can be upper bounded by the gradient variation $V_T$ and the summation of $\|\mathbf{x}_t - \bar{\mathbf{x}}_t\|_2^2$. The latter one can be further expanded as

$$\|\mathbf{x}_t - \bar{\mathbf{x}}_t\|_2^2 = \Big\|\sum_{i=1}^{N}(p_{t,i} - p_{t-1,i})\mathbf{x}_{t,i}\Big\|_2^2 \leq \Big(\sum_{i=1}^{N}|p_{t,i} - p_{t-1,i}|\|\mathbf{x}_{t,i}\|_2\Big)^2 \leq D^2\|\boldsymbol{p}_t - \boldsymbol{p}_{t-1}\|_1^2,$$

which can be eliminated by the negative term in (8), with a suitable setting of the learning rate $\varepsilon$.

### 2.2.3 Regret Guarantees

We prove that the meta-regret of VariationHedge is $\mathcal{O}(\sqrt{V_T \ln N})$, compatible to the expert-regret.

**Theorem 2.** *Under Assumptions 1, 2, and 3, by setting the learning rate optimally as* $\varepsilon = \min\{\sqrt{1/(8D^4 L^2)}, \sqrt{(2 + \ln N)/(2D^2 V_T)}\}$, *the meta-regret of VariationHedge is at most*

$$\texttt{meta-regret} \leq 2D\sqrt{2V_T(2 + \ln N)} + 4\sqrt{2}D^2 L(2 + \ln N) = \mathcal{O}(\sqrt{V_T \ln N}).$$

Note that the dependence on $V_T$ in the optimal learning rate tuning can be removed by the doubling trick. Furthermore, actually we can set the optimal learning rate of the meta-algorithm with $\widehat{V}_T = \sum_{t=2}^{T}\|\nabla f_t(\mathbf{x}_t) - \nabla f_{t-1}(\mathbf{x}_t)\|_2^2$ instead of the original gradient variation $V_T$ via a more refined analysis. The quantity $\widehat{V}_T$ can be regarded as an empirical approximation of $V_T$, and it can be calculated directly without involving the inner problem of $\sup_{\mathbf{x}\in\mathcal{X}}\|\nabla f_t(\mathbf{x}) - \nabla f_{t-1}(\mathbf{x})\|_2^2$. Thereby, we can perform the doubling trick by monitoring $\widehat{V}_T$ with much less computational efforts. Combining Theorem 1 (expert-regret) and Theorem 2 (meta-regret), we have the following dynamic regret bound.

**Theorem 3.** *Under Assumptions 1, 2, and 3, setting the pool of candidate step sizes $\mathcal{H}_{var}$ as*

$$\mathcal{H}_{var} = \left\{\eta_i = 2^{i-1}\sqrt{\frac{D^2}{2GT}}, i \in [N_1]\right\}, \qquad (11)$$

*where $N_1 = \lceil 2^{-1} \log_2(GT/(8D^2L^2)) \rceil + 1$.[1] Then Sword$_{var}$ (Algorithms 1 and 2) satisfies*

$$\sum_{t=1}^{T} f_t(\mathbf{x}_t) - \sum_{t=1}^{T} f_t(\mathbf{u}_t) \leq \mathcal{O}\Big( \sqrt{(1 + P_T + V_T)(1 + P_T)} \Big)$$

*for any comparator sequence $\mathbf{u}_1, \ldots, \mathbf{u}_T \in \mathcal{X}$.*

**Remark 2.** Compared with the existing $\mathcal{O}(\sqrt{T(1 + P_T)})$ dynamic regret [8], our result is more adaptive in the sense that it replaces $T$ by the *problem-dependent* quantity $P_T + V_T$. Therefore, the bound will be much tighter in easy problems, for example when both $V_T$ and $P_T$ are $o(T)$. Meanwhile, it safeguards the same minimax rate, since both quantities are at most $\mathcal{O}(T)$.

**Remark 3.** Because the *universal* dynamic regret studied in this paper holds against any comparator sequence, it specializes the static regret by setting all comparators as the best fixed decision in hindsight, i.e., $\mathbf{u}_1 = \ldots = \mathbf{u}_T = \mathbf{x}^* \in \arg\min_{\mathbf{x} \in \mathcal{X}} \sum_{t=1}^{T} f_t(\mathbf{x})$. Under such a circumstance, the path-length $P_T = \sum_{t=2}^{T} \|\mathbf{u}_{t-1} - \mathbf{u}_t\|_2$ will be zero, so the regret bound in Theorem 3 actually implies an $\mathcal{O}(\sqrt{V_T})$ variation static regret bound, which recovers the result of Chiang et al. [15].

## 2.3 Small-Loss Bound

In this part, we turn to another problem-dependent quantity, cumulative loss of the comparator sequence, and prove the small-loss dynamic regret. We start from the online gradient descent (OGD),

$$\mathbf{x}_{t+1} = \Pi_{\mathcal{X}}\big[\mathbf{x}_t - \eta \nabla f_t(\mathbf{x}_t)\big]. \tag{12}$$

Srebro et al. [14] prove that OGD achieves an $\mathcal{O}(\sqrt{F_T^*})$ static regret, where $F_T^* = \sum_{t=1}^{T} f_t(\mathbf{x}^*)$ is the cumulative loss of the comparator benchmark $\mathbf{x}^*$. For the dynamic regret, since the benchmark is changing, a natural replacement is the cumulative loss of the comparator sequence $\mathbf{u}_1, \ldots, \mathbf{u}_T$, namely $F_T = \sum_{t=1}^{T} f_t(\mathbf{u}_t)$. We show that OGD indeed enjoys such a small-loss dynamic regret.

**Theorem 4.** *Under Assumptions 2, 3, and 4, by choosing any step size $\eta \leq \frac{1}{4L}$, OGD satisfies*

$$\sum_{t=1}^{T} f_t(\mathbf{x}_t) - \sum_{t=1}^{T} f_t(\mathbf{u}_t) \leq \frac{D^2 + 2DP_T}{2\eta(1 - 2\eta L)} + \frac{2\eta L}{1 - 2\eta L} \sum_{t=1}^{T} f_t(\mathbf{u}_t) = \mathcal{O}\Big( \frac{1 + P_T}{\eta} + \eta F_T \Big)$$

*for any comparator sequence $\mathbf{u}_1, \ldots, \mathbf{u}_T \in \mathcal{X}$.*

Similar to Sword$_{var}$, the step size needs to balance between non-stationarity ($P_T$) and adaptivity ($F_T$, this time). Notice that the optimal tuning depends on $P_T$ and $F_T$, both of which are unknown even after all $T$ iterations. Therefore, we again compensate the lack of this information via the meta-expert framework to hedge the non-stationarity while keeping the adaptivity. The expert-algorithm is set as OGD. The meta-algorithm is required to suffer a small-loss meta-regret of order $\mathcal{O}(\sqrt{F_T \ln N})$. We discover that vanilla Hedge with linearized surrogate loss is qualified, which updates the weight by

$$p_{t+1,i} \propto \exp\Big( -\varepsilon \sum_{s=1}^{t} \langle \nabla f_s(\mathbf{x}_s), \mathbf{x}_{s,i} \rangle \Big), \quad \forall i \in [N]. \tag{13}$$

Notice that vanilla Hedge can be treated as OptimisticHedge with null optimism, i.e., $\boldsymbol{m}_{t+1} = \mathbf{0}$. Therefore, by Lemma 1 we know that its meta-regret is of order $\mathcal{O}(\sqrt{D_\infty \ln N})$ and

$$D_\infty = \sum_{t=1}^{T} \max_{i \in [N]} \langle \nabla f_t(\mathbf{x}_t), \mathbf{x}_{t,i} \rangle^2 \leq D^2 \sum_{t=1}^{T} \|\nabla f_t(\mathbf{x}_t)\|_2^2 \leq 4D^2 L \sum_{t=1}^{T} f_t(\mathbf{x}_t), \tag{14}$$

where the last inequality follows from the self-bounding property of smooth functions [14, Lemma 3.1]. As a result, the meta-regret is now $\mathcal{O}(\sqrt{F_T^{\mathbf{x}} \ln N})$, where $F_T^{\mathbf{x}} = \sum_{t=1}^{T} f_t(\mathbf{x}_t)$ is the cumulative loss of decisions. Note that the term $F_T^{\mathbf{x}}$ can be further processed to the desired small-loss quantity $F_T = \sum_{t=1}^{T} f_t(\mathbf{u}_t)$, the cumulative loss of comparators. We will present details in the proof.

To summarize, Sword$_{small}$ chooses OGD (12) as the expert-algorithm, and uses the vanilla Hedge with linearized surrogate loss as the meta-algorithm shown in the update form (13). The theorem below shows that the proposed algorithm enjoys the small-loss dynamic regret bound.

**Theorem 5.** *Under Assumptions 1, 2, 3, and 4, setting the pool of candidate step sizes* $\mathcal{H}_{small}$ *as*

$$\mathcal{H}_{small} = \left\{ \eta_i = 2^{i-1}\sqrt{\frac{D}{16LGT}}, i \in [N_2] \right\}, \tag{15}$$

*where* $N_2 = \lceil 2^{-1}\log_2(GT/(DL)) \rceil + 1$. *Setting the learning rate of meta-algorithm optimally as* $\varepsilon = \sqrt{(2 + \ln N_2)/(D^2 F_T^{\mathbf{x}})}$, *then* $Sword_{small}$ *satisfies*

$$\sum_{t=1}^{T} f_t(\mathbf{x}_t) - \sum_{t=1}^{T} f_t(\mathbf{u}_t) \leq \mathcal{O}\left(\sqrt{(1 + P_T + F_T)(1 + P_T)}\right).$$

*for any comparator sequence* $\mathbf{u}_1, \dots, \mathbf{u}_T \in \mathcal{X}$.

Note that the optimal learning rate tuning requires the knowledge of $F_T^{\mathbf{x}}$, which can be easily removed by doubling trick or self-confident tuning [25], since it is empirically evaluable at each iteration. Moreover, the $\mathcal{O}(\sqrt{(1 + P_T + F_T)(1 + P_T)})$ universal dynamic regret in Theorem 5 specializes to the $\mathcal{O}(\sqrt{F_T})$ static regret [14] when setting the comparators as the fixed best decision in hindsight.

## 3 Best-of-Both-Worlds Bound

In the last section, we propose $Sword_{var}$ and $Sword_{small}$ that achieve variation and small-loss bounds respectively. Due to different problem-dependent quantities are involved, these two bounds are generally incomparable and are favored in different scenarios. Therefore, it is natural to ask for a *best-of-both-worlds* guarantee: the regret of the minimum of variation and small-loss bounds.

To this end, we require a meta-algorithm that can enjoy both kinds of adaptivity to combine all the experts, with an $\mathcal{O}(\sqrt{\min\{V_T, F_T\}\ln N})$ meta-regret. Based on the observation that *both VariationHedge and vanilla Hedge are essentially special cases of OptimisticHedge with different configurations of optimism*,

Table 1: Summary of expert-algorithms and meta-algorithms as well as different optimism used in the proposed algorithms (including three variants of Sword).

| Method | Expert | Meta | Optimism |
|---|---|---|---|
| $Sword_{var}$ | OEGD | VariationHedge | by (9) |
| $Sword_{small}$ | OGD | vanilla Hedge | $\boldsymbol{m}_{t+1} = \mathbf{0}$ |
| $Sword_{best}$ | OEGD & OGD | OptimisticHedge | by (18), (21) |

we adopt the OptimisticHedge to be the meta-algorithm for $Sword_{best}$, where a parallel meta-algorithm is introduced to *learn the best optimism* for OptimisticHedge to ensure best-of-both-worlds meta-regret. In the following we describe the expert-algorithm and meta-algorithm of $Sword_{best}$.

**Expert-algorithm.** We aggregate the experts of $Sword_{var}$ and $Sword_{small}$, so there are $N = N_1 + N_2$ experts in total and the step size of each experts is set according to the pool $\mathcal{H} = \mathcal{H}_{var} \cup \mathcal{H}_{small}$ (cf. (11) and (15) for definitions). The first $N_1$ experts run OEGD (6) with the step size chosen from $\mathcal{H}_{var}$, and the other $N_2$ experts perform OGD (12) with step size specified by $\mathcal{H}_{small}$. At iteration $t$, the final output is a weighted combination of predictions returned by the expert-algorithms, namely,

$$\mathbf{x}_t = \sum_{i=1}^{N} p_{t,i}\mathbf{x}_{t,i} = \sum_{i=1}^{N_1} p_{t,i}\mathbf{x}_{t,i}^{v} + \sum_{i=N_1+1}^{N_1+N_2} p_{t,i}\mathbf{x}_{t,i}^{s}, \tag{16}$$

where $\boldsymbol{p}_t \in \Delta_{N_1+N_2}$ is the weight, $\mathbf{x}_{t,i} = \mathbf{x}_{t,i}^{v}$ for $i = 1, \dots, N_1$ are predictions returned by the expert-algorithms (OEGD) of $Sword_{var}$, and $\mathbf{x}_{t,i} = \mathbf{x}_{t,i}^{s}$ for $i = N_1 + 1, \dots, N_1 + N_2$ are predictions returned by the expert-algorithms (OGD) of $Sword_{small}$. It remains to specify the meta-algorithm.

**Meta-algorithm.** We adopt the OptimisticHedge algorithm along with the linearized surrogate loss as the meta-algorithm, where the weight vector $\boldsymbol{p}_{t+1} \in \Delta_{N_1+N_2}$ is updated according to

$$p_{t+1,i} \propto \exp\left(-\varepsilon\left(\sum_{s=1}^{t}\langle\nabla f_s(\mathbf{x}_s), \mathbf{x}_{s,i}\rangle + m_{t+1,i}\right)\right), \tag{17}$$

where the optimism $\boldsymbol{m}_{t+1} \in \mathbb{R}^{N_1+N_2}$. In order to facilitate the meta-algorithm with both kinds of adaptivity ($V_T$ and $F_T$), it is crucial to design best-of-both-worlds optimism.

| **Algorithm 3** Sword$_{\text{best}}$: Meta (OptimisticHedge) | **Algorithm 4** Sword$_{\text{best}}$: Expert (OEGD & OGD) |
|---|---|
| **Input:** step size pool $\mathcal{H}$; learning rate $\varepsilon$ | **Input:** step size $\eta_i$ |
| 1: Let $N = N_1 + N_2$, $\forall i \in [N]$, $p_{0,i} = 1/N$ | 1: Let $\hat{\mathbf{x}}_{1,i}, \mathbf{x}_{1,i}$ be any point in $\mathcal{X}$ |
| 2: **for** $t = 1$ **to** $T$ **do** | 2: **for** $t = 1$ **to** $T$ **do** |
| 3:　　Receive prediction $\mathbf{x}_{t+1,i}$ from expert $\mathcal{E}_i$ | 3:　　**if** $i \in \{1, \ldots, N_1\}$ **then** |
| 4:　　Set $M^v_{t+1}$ and $M^s_{t+1}$ by (19) and (20) | 4:　　　$\hat{\mathbf{x}}_{t+1,i} = \Pi_{\mathcal{X}}\left[\hat{\mathbf{x}}_{t,i} - \eta_i \nabla f_t(\mathbf{x}_{t,i})\right]$ |
| 5:　　Update the weight $\beta_{t+1}$ by (22) | 5:　　　$\mathbf{x}_{t+1,i} = \Pi_{\mathcal{X}}\left[\hat{\mathbf{x}}_{t+1,i} - \eta_i \nabla f_t(\hat{\mathbf{x}}_{t+1,i})\right]$. |
| 6:　　Obtain the optimism $M_{t+1}$ (21) | 6:　　**else** |
| 7:　　Update the weight $p_{t+1,i}$ by (17) and (18) | 7:　　　$\mathbf{x}_{t+1,i} = \Pi_{\mathcal{X}}\left[\mathbf{x}_{t,i} - \eta_i \nabla f_t(\mathbf{x}_{t,i})\right]$. |
| 8:　　Output the prediction | 8:　　**end if** |
| 　　　$\mathbf{x}_{t+1} = \sum_{i=1}^{N} p_{t+1,i}\mathbf{x}_{t+1,i}$ | 9:　　Send prediction $\mathbf{x}_{t+1,i}$ to meta-algorithm |
| 9: **end for** | 10: **end for** |

We set the optimism $\boldsymbol{m}_{t+1}$ in the following way: for each $i \in [N_1 + N_2]$

$$m_{t+1,i} = \langle M_{t+1}, \mathbf{x}_{t+1,i}\rangle, \tag{18}$$

where $M_{t+1} \in \mathbb{R}^d$ is called the optimistic vector. So we are left with the task of determining the term of $M_{t+1}$ in (18). Inspired by the seminal work of Rakhlin and Sridharan [16], we treat the problem of selecting the sequence of optimistic vectors as another online learning problem. The idea is to build a parallel meta-algorithm for learning the optimistic vector $M_{t+1}$, which is then fed to OptimisticHedge of (17) for combining multiple experts, to achieve a best-of-both-worlds meta-regret.

Specifically, consider the following learning scenario of *prediction with two expert advice*. At the beginning of iteration $(t + 1)$, we receive two optimistic vectors $M^v_{t+1}, M^s_{t+1} \in \mathbb{R}^d$, based on which the algorithm determines the optimistic vector $M_{t+1} \in \mathbb{R}^d$ for Sword$_{\text{best}}$. Then the online function $f_{t+1}$ is revealed, and we subsequently observe the loss of $d_{t+1}(M^v_{t+1})$ and $d_{t+1}(M^s_{t+1})$, where $d_{t+1}(M) = \|\nabla f_{t+1}(\mathbf{x}_{t+1}) - M\|^2_2$. In above, the vectors of $M^v_{t+1}$ and $M^s_{t+1}$ are

$$M^v_{t+1} = \nabla f_t(\bar{\mathbf{x}}_{t+1}), \quad \text{and} \quad M^s_{t+1} = \mathbf{0}, \tag{19}$$

where $\bar{\mathbf{x}}_{t+1}$ is the instrumental output. Similar to the construction of (9), it is designed as

$$\bar{\mathbf{x}}_{t+1} = \sum_{i=1}^{N_1} p_{t,i}\mathbf{x}^v_{t+1,i} + \sum_{i=N_1+1}^{N_1+N_2} p_{t,i}\mathbf{x}^s_{t+1,i}. \tag{20}$$

Notice that the function $d_t : \mathbb{R}^d \mapsto \mathbb{R}$ is 2-strongly convex with respect to $\|\cdot\|_2$-norm, we thus choose Hedge of strongly convex functions [26, Chapter 3.3] as the parallel meta-algorithm for updating,

$$M_{t+1} = \beta_{t+1}M^v_{t+1} + (1 - \beta_{t+1})M^s_{t+1}, \tag{21}$$

where the weight $\beta_{t+1} \in [0, 1]$ for learning optimistic vectors is updated by

$$\beta_{t+1} = \frac{\exp(-2D^v_t)}{\exp(-2D^v_t) + \exp(-2D^s_t)} \tag{22}$$

with $D^v_t = \sum_{\tau=1}^t d_\tau(M^v_\tau)$ and $D^s_t = \sum_{\tau=1}^t d_\tau(M^s_\tau)$.

Algorithm 3 summarizes the meta-algorithm of Sword$_{\text{best}}$. In the last two columns of Table 1, we present comparisons of the meta-algorithms and optimism designed for different methods .

**Regret Analysis.** Recall that the meta-regret of OptimisticHedge is of order $\mathcal{O}(\sqrt{D_\infty \ln N})$. From the setting of surrogate loss (17) and optimism (18), we have

$$D_\infty = \sum_{t=1}^T \max_{i\in[N]} \left(\langle \nabla f_t(\mathbf{x}_t) - M_t, \mathbf{x}_{t,i}\rangle\right)^2 \le D^2 \sum_{t=1}^T \|\nabla f_t(\mathbf{x}_t) - M_t\|^2_2.$$

Besides, the regret analysis of Hedge for strongly convex functions [26, Proposition 3.1] implies

$$\sum_{t=1}^T \|\nabla f_t(\mathbf{x}_t) - M_t\|^2_2 = \sum_{t=1}^T d_t(M_t) \le \min\left\{\bar{V}_T, \bar{F}_T\right\} + \frac{\ln 2}{2},$$

where $\bar{V}_T = \sum_{t=2}^T \|\nabla f_t(\mathbf{x}_t) - \nabla f_{t-1}(\bar{\mathbf{x}}_t)\|^2_2$ and $\bar{F}_T = \sum_{t=1}^T \|\nabla f_t(\mathbf{x}_t)\|^2_2$. The two terms can be further converted to the desired gradient variation $V_T$ and small loss $F_T$, by exploiting the smoothness and expert-regret analysis. We can thus ensure the following meta-regret bound.

**Theorem 6.** *Under Assumptions 1, 2, 3, and 4, by setting the learning rate optimally as $\varepsilon = \min\{\sqrt{1/(8D^4L^2)}, \varepsilon^*\}$, the meta-algorithm of Sword$_{best}$ satisfies*

$$\texttt{meta-regret} \leq 2D\sqrt{(2+\ln N)(\min\{2V_T, \bar{F}_T\} + \ln 2)} + 4\sqrt{2}D^2L(2+\ln N)$$

*where $\varepsilon^* = \sqrt{(2+\ln N)/(D^2\min\{2V_T, \bar{F}_T\} + D^2\ln 2)}$.*

Because $V_T$ and $\bar{F}_T$ are both empirically observable, we can easily get rid of their dependence in the optimal learning rate tuning. Also see the discussion below Theorem 2 about replacing the original gradient variation $V_T$ by its empirical approximation $\widehat{V}_T = \sum_{t=2}^{T}\|\nabla f_t(\mathbf{x}_t) - \nabla f_{t-1}(\mathbf{x}_t)\|_2^2$ to save computational costs. Besides, the $\bar{F}_T$ term of meta-regret will be converted to the desired small-loss quantity $F_T$ in the final regret bound. Combining above meta-regret analysis as well as the expert-regret analysis of OEGD and OGD algorithms, we can finally achieve the best of both worlds.

**Theorem 7.** *Under Assumptions 1, 2, 3, and 4, setting the pool of candidate step sizes as*

$$\mathcal{H} = \mathcal{H}_{var} \cup \mathcal{H}_{small}, \tag{23}$$

*where $\mathcal{H}_{var}$ and $\mathcal{H}_{small}$ are defined in (11) and (15). Then Sword$_{best}$ (Algorithms 3 and 4) satisfies*

$$\sum_{t=1}^{T} f_t(\mathbf{x}_t) - \sum_{t=1}^{T} f_t(\mathbf{u}_t) \leq \mathcal{O}\big(\sqrt{(1 + P_T + \min\{V_T, F_T\})(1 + P_T)}\big),$$

*for any comparator sequence $\mathbf{u}_1, \ldots, \mathbf{u}_T \in \mathcal{X}$.*

**Remark 4.** The dynamic regret bound in Theorem 7 achieves a minimum of gradient-variation and small-loss bounds, and therefore combines their advantages and enjoys both kinds of adaptivity. Moreover, the result also implies an $\mathcal{O}(\sqrt{\min\{V_T, F_T\}})$ static regret by setting the sequence of comparators to be the best fixed decision in hindsight, where we note that $F_T$ is now the same as the notation of $F_T^*$ used below (12), the cumulative loss of the comparators benchmark.

## 4 Conclusion

In this paper, we exploit smoothness to enhance the universal dynamic regret, with the aim to replace the time horizon $T$ in the state-of-the-art $\mathcal{O}(\sqrt{T(1 + P_T)})$ bound by *problem-dependent* quantities that are at most $\mathcal{O}(T)$ but can be much smaller in easy problems. We achieve this goal by proposing two meta-expert algorithms: Sword$_{var}$ which attains a gradient-variation dynamic regret bound of order $\mathcal{O}(\sqrt{(1 + P_T + V_T)(1 + P_T)})$, and Sword$_{small}$ which enjoys a small-loss dynamic regret bound of order $\mathcal{O}(\sqrt{(1 + P_T + F_T)(1 + P_T)})$. Here, $V_T$ measures the variation in gradients and $F_T$ is the cumulative loss of the comparator sequence. They are at most $\mathcal{O}(T)$ yet could be very small when the problem is easy, and thus reflect the difficulty of the problem instance. As a result, our bounds improve the minimax rate of universal dynamic regret by exploiting smoothness. Finally, we design Sword$_{best}$ to combine advantages of both gradient-variation and small-loss algorithms and achieve a best-of-both-worlds dynamic regret bound of order $\mathcal{O}(\sqrt{(1 + P_T + \min\{V_T, F_T\})(1 + P_T)})$. We note that all of attained dynamic regret bounds are universal in the sense that they hold against *any* feasible comparator sequence, and thus the algorithms are more adaptive to the non-stationary environments. In the future, we will investigate the possibility of exploiting other function curvatures, such as strong convexity or exp-concavity, into the analysis of the universal dynamic regret.

## Acknowledgment

This research was supported by the National Key R&D Program of China (2018YFB1004300), NSFC (61921006, 61976112), the Collaborative Innovation Center of Novel Software Technology and Industrialization, and the Baidu Scholarship. The authors would like to thank Mengxiao Zhang for helpful discussions. We are also grateful for the anonymous reviewers for their valuable comments.

## Broader Impact

The work is mostly theoretical and the broader impact discussion is not applicable.

## Footnotes

[1]The number of candidate step sizes is denoted by $N_1$ instead of $N$ to distinguish it with that of Sword$_{small}$.

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
