[Reviews · NeurIPS 2020]

Review 1

Summary and Contributions: This paper provides algorithms for online convex optimization with smooth non-negative losses that achieve dynamic regret sqrt( P^2 + P min (F, V) ) where P is the path length of the comparator sequence, F is the total loss of the comparator sequence, and V is a measure of variation in the gradients of the losses. The minimax regret for this setting is sqrt(TP), and the authors show that a regret of P is unavoidable. The main technique is the by-now-standard approach for dynamic regret: instantiate many base learners and aggregate them using some meta-learner. In this case, the difficulty is to 1: make sure the best base learner indeed obtains the desired bound, and 2: makes sure the aggregator algorithm does not destroy this property. Both are achieved by various appropriate invocations of optimistic online learning or the self-bounding principle for smooth losses.

Strengths: So far as I know this is a new result, while not in my opinion extremely surprisingly, and the analysis requires some work to complete. The paper motivates the problem well. The lower bound for regret at least P is nice to have.

Weaknesses: I think this paper would be much stronger if it could provide some technique other than combining all the adaptive bounds together and verifying that nothing breaks. Is there any hope to avoid the log(T) blowup in runtime? Is it at least possible to avoid having to aggregate all experts from both algorithms to get the desired best-of-both worlds bound? (maybe by getting the variation V to be replaced with ||\nabla f_t(x_t)- \nabla f_{t-1}(x_t)\|^2 rather than a sup over all x?) Technically, I think in order for Lemma 4 to hold, f needs to be defined on the whole vector space rather than just the compact set W (e.g. otherwise the result can be easily violated by an affine function: consider f(x) = x+1 on [-1,1]). ------- I have read the response and the other reviews. The authors suggested some new experimental results, and suggested a way to get some kind of combined regret bound albeit with a slightly different definition of V_T. Based on this, and further reflection on the novelty issues, I will increase my score by one point.

Correctness: I believe so.

Clarity: yes

Relation to Prior Work: yes

Reproducibility: Yes

Additional Feedback:


Review 2

Summary and Contributions: The authors examine the question of whether smoothness can help improve dynamic regret bounds in online convex optimization. In particular, they ask whether smoothness can be used to remove existing dependencies on the problem horizon T by other natural problem parameters, such as the gradient variation or cumulative loss of the comparator sequence, which can be small in cases where the relevant sequence of functions changes slowly. Contrary to other work, which has already studied the possibility of improved worst case dynamic regret bounds in OCO when the objectives are smooth, this paper analyzes whether smoothness can help improve regret against an arbitrary sequence of comparators (not just the sequence of optimal solutions). They draw upon techniques from the online learning community to adapt existing algorithms, such as Online Extra Gradient Descent, and Hedge (MWU) to this setting. Their main result is a new algorithm, which they term Sword, which achieves a dynamic regret bound with no explicit dependence on the problem horizon, but instead depends on problem specific quantities.

Strengths: As far as I can tell, these dynamic regret bounds with respect to an arbitrary sequence of comparators are new and they neatly reveal how upper bounds on dynamic regret can be small in cases where the sequence of comparators is slowly changing. On a technical level, the authors provide an interesting construction of a new online learning algorithm that brings together key ideas from various existing procedures.

Weaknesses: In my view, the paper could be significantly improved by spending more time at the beginning explaining why bounds on universal dynamic regret (where the sequence of comparators is arbitrary) would be preferred over worst case dynamic regret bounds. It seems like the overall value of the paper hinges on making a strong argument here, since the effects of smoothness are already understood when the goal is to prove a worst-case dynamic regret bound. Any connections to related work showing how improving universal regret bounds improves our understanding of problems in related areas would be much appreciated. Furthermore, at a technical level, I was left wondering what regret guarantees are possible if the learner only has access to 1 gradient query per step, rather than the two used in OEGD.

Correctness: Due to time constraints, I was only able to make high level checks of the relevant proofs.

Clarity: Yes, the author’s writing is clear.

Relation to Prior Work: Yes, the authors do a reasonable job of situating their results within the broader literature.

Reproducibility: Yes

Additional Feedback: Minor comments: The authors clearly mention that they treat double logarithmic factors in T as constant (as done in previous work), but at the same time it seems strange to me to leave those out of the relevant theoretical bounds if part of the goal of the paper is to remove dependencies on the time horizon. This is only a minor stylistic point. ======= Update ======= Thanks for the detailed answered to my questions. I'm still not fully convinced that there are other comparator sequences other than worst case regret which are worth comparing to.


Review 3

Summary and Contributions: After reading the author response: -------- I have read the author response and the other reviews, and the paper still ranks high in my view. Here are my comments: - "Can we avoid meta-grad"? I'm not sure how, given the presence of $P_T$. I would have suggested something like AdaGrad step-sizes for adapting to $V_T$ without multiple copies and the doubling trick, e.g.: [Joulani et al., TCS-2020, "A Modular Analysis of Adaptive (Non-)Convex Optimization...", Corollary 9 /Table 4] but the issue is we cannot measure $P_T$ when the dynamic regret comes in. - "Still, it's just meta-grad" / "novelty in the analysis": I agree that the paper heavily relies on meta-grad, and a more interesting result would be something that avoids the excessive copying of experts. However, I still think this is a nice first step, as the paper is clearly distilling the challenges involved (which is useful for others who want to build on this work) and then overcoming those challenges in a meaningful, non-trivial way. In particular: + The design of the optimistic prediction vector (8) for the first set of bounds is indeed nice, and the need for it is clearly explained. + The different algorithmic choices they make are somewhat inevitable, given their definition of $V_T$ and $F_T$ (e.g., they indeed have to use a non-optimistic algorithm for the second set of bounds, etc.). I agree with their response to the question by Reviewer 1: if we change the definition of $V_T$, obtaining the best-of-both-worlds bound seems straightforward. - "universal vs worst-case dynamic regret": I agree with the authors that, while the worst-case dynamic regret bound might seem stronger, since we are comparing with a stronger competitor there, having a strong target makes all algorithms equally incapable, going against what we typically want to achieve in these "adaptive" bounds. My understanding is that by "less adaptive", the authors actually mean "less flexible": what if we have a slow-changing sequence of competitors? With a bound that depends on $P_T$, we can be sure that the algorithm will perform well. With a worst-case bound, we won't be able to make that assertion. - "why are comparator sequences other than worst case regret useful?": This is especially useful in applications in which any practical solution must change slowly in time or there are, e.g., constraints on the set from which the comparator at a point has to be chosen (that is, coming up with the worst-case competitor is not desirable due to the problem constraints, hence comparing to it is not useful). For example, when coming up with dynamic web content, one may not want to use an old article as the main, top-page hit, even if it is more likely to be clicked on (one could also come up with better examples, I imagine :) ) -------- The paper provides new online learning algorithms with improved dynamic regret when the loss functions underlying the problem are smooth. In particular, the new algorithms enjoy bounds that depend on: - P_T: the path-length of the comparator sequence - V_T: the total gradient variation in the loss sequence - F_T: the total loss of the comparator sequence Three algorithms are proposed. Each algorithm replaces $T$ (time-horizon) in the dynamic regret bound of the algorithm of Zhang et al. [10] by a quantity that is at most O(T) (hence preserving the worst-case bound) but that can in simple, easy games be much smaller than the worst-case $T$. In particular: 1) The first algorithm, SWORD_var, replaces the $T$ by (1+P_T+V_T) 2) The second algorithm, SWORD_small, replaces $T$ by (1+P_T+F_T) 3) The third algorithm, SWORD_best, combiens the two other algorithms, replacing $T$ with (1+P_T + min{V_T,F_T}). The main technical challenge, that is effectively overcome, is how to use a proper optimistic prediction for the OptimisticHedge algorithm that is used as the meta-learner in a meta-grad scheme (used to avoid tuning the step-size based on the unknown value of P_T). The authors further provide a lower bound demonstrating that the presence of $P_T$ in the quantities above is inevitable.

Strengths: The paper is, in general, of high quality. The setting is layed out clearly, the algorithms are motivated adequately, and the analysis flows smoothly. The arguments are convincing and point in the right direction; the approach the paper takes to overcome the technical challenges makes sense and is well carried out. The results will be interesting to the online learning community, especially given the new interest in dynamic regret and meta-gradient algorithms.

Weaknesses: There is a minor problem with the assumptions made (i.e., slightly stronger assumptions are required for the off-the-shelf results the author use). The other potential problem is the high cost of using a meta-grad algorithm. See question 8 of the review for further details.

Correctness: The theory flows in the right direction, and the results seem correct in general. There are some details of the proofs that I have not checked, but I plan to check them so I would like to ask the authors to do a double-check of the proofs, and provide an errata in the author response if applicable.

Clarity: The paper is written very clearly, and is easy to follow, at least for those with prior exposure to adaptive bounds for online learning.

Relation to Prior Work: The paper provides an appendix that further discusses the relation to previous work. While the information is there, it may still be useful to provide a table that puts the non-dynamic as well as dynamic but non-adaptive previous work and the form of their bounds side by side, so that the contributions are better appreciated.

Reproducibility: Yes

Additional Feedback: Self-bounding property of smooth functions: ------------------------------------------- Assumption 3 does not seem to imply the self-bounding property of the functions, even when combined with $f_t$ being non-negative. Upon inspecting the proof of [11, Lemma 3.1], one can observe that in the second inequality of their proof, the authors indeed use Assumption 3 holding over the whole $R$ (since $t,s$ are real values, $w_0 + t(w-w_0)$ will, in general, lie outside any compact set $W$ for large enough $t$. Hence, for the second inequality of the proof of [11] to follow, Assumption 3 must hold outside $X$ as well). The problem is resolved if Assumption 3 is modified to hold over the whole $R^d$ rather than only on $X$. Questions: ---------- - Q1: Are Assumptions 3 and/or 4 without loss of (much) generality? In particular, how would the lower-bound of function appear in your bounds if we assume they are not positive, and how would the algorithms / results change if $0$ is not in $X$? Other comments, typos, etc.: ---------------------------- - Line 37: in the definition of $P_T$, sum must start from $t=2$ (or $u_0 := u_1$ be defined) - Line 34: "is more adaptive to the non-stationary...": perhaps consider rephrasing, I guess "more flexible to use with non-stationary..." is more inline with what you meant? - Perhaps it's useful to elaborate a bit why you need "null optimism" for $Sword_{small}$, in particular, because we need the gradients to have been taken at the exact iterates of that time step, otherwise we will not have $F_t$. - One suggestion is to consider using the $\tilde{O}$ notations rather than "treat double logarithmic factors in T as constant". - Suggestion: Line 96: "also enjoys the [...] regret" -> "enjoys a dynamic regret bound" instead? - Line 190 / Theorem 5: It seems you still need Assumption 1 for $G$ to be defined.


Review 4

Summary and Contributions: The paper deals with online convex optimization in non-stationary environments while focusing on dynamic regret as a performance measure. In this paper, the authors exploited the smoothness condition of the loss function to improve the dynamic regret measure. The above is achieved by replacing the dependence on T in the dynamic regret by the variation in gradients of loss functions.

Strengths: New algorithms that exploits the smoothness of the loss function.

Weaknesses: In my opinion, such a paper must demonstrate the results on several tasks (even on toy problems). The fact that no experiments exist hurt the evaluation of this work. The novelty of the paper is limited. The algorithm is a combination of known and well-studied ideas. I do except such theoretical paper to present some sort of novelty in the analysis. ------- After reading the response and the other reviews. I was pleased by the answers provided by the authors and therefore I raised my score

Correctness: I didn't find any flaws.

Clarity: Yes

Relation to Prior Work: Yes.

Reproducibility: No

Additional Feedback:

[Author Response · NeurIPS 2020]

**[To Reviewer #1]** Thanks for your constructive comments. We answer your main questions as follows.

**Question 1.** "Is there any hope to avoid the $\log(T)$ blowup in runtime? Is it at least possible to avoid having to

aggregate all experts from both algorithms to get the desired best-of-both worlds bound?"

**Answer 1.** From our current understanding, the meta-expert aggregation is the standard framework for handling the

uncertainty of non-stationary online learning. For your second question, yes, we can design an algorithm that avoids

aggregating all experts from both algorithms to achieve $\mathcal{O}(\sqrt{(1 + P_T + \min\{\bar{V}_T, F_T\})(1 + P_T)})$ dynamic regret,

where $\bar{V}_T = \sum_{t=2}^{T}\|\nabla f_t(\mathbf{x}_t) - \nabla f_{t-1}(\mathbf{x}_{t-1})\|^2$. To do so, we require the expert-algorithm to run *optimistic gradient*

*descent* with the learned optimism. We will add a remark in the paper to discuss this point more thoroughly.

**Question 2.** "Technically, I think in order for Lemma 4 to hold, $f$ needs to be defined on the whole vector space"

**Answer 2.** Thanks for your comments! You are correct. The issue has also been identified by Reviewer #3. We can

resolve it by requiring $f_t$ to be defined on $\mathbb{R}^d$, and we will correct it in the revised version.

---

**[To Reviewer #2]** Thanks for your helpful comments, and we address your concerns as follows.

**Question 1.** about improving paper by spending more efforts in explaining the advantages of universal dynamic regret

**Answer 1.** Thanks for your suggestion. Universal dynamic regret supports arbitrary comparator sequence, and thus it

subsumes the worst-case dynamic regret and static regret as special cases. So it is a more adaptive performance measure

for non-stationary online learning. We will improve the paper writing to make this point more clear.

**Question 2.** "what regret ... if ... only access to 1 gradient query per step, rather than the two used in OEGD."

**Answer 2.** If the expert-algorithm only has access to 1 gradient per step, $\mathcal{O}(\sqrt{(1 + P_T + \bar{V}_T)(1 + P_T)})$ dynamic

regret is attainable, where $\bar{V}_T = \sum_{t=2}^{T}\|\nabla f_t(\mathbf{x}_t) - \nabla f_{t-1}(\mathbf{x}_{t-1})\|^2$ is slightly different from the gradient-variation $V_T$:

$V_T$ takes the sup over all $\mathbf{x}$ and thus is fully problem-dependent; while $\bar{V}_T$ will depend on the decision path $\{\mathbf{x}_t\}_{t=1}^{T}$.

---

**[To Reviewer #3]** Thanks for your appreciation and insightful review! We address your main questions as follows.

**Question 1.** "how would the lower-bound of function appear in your bounds if we assume they are not positive"

**Answer 1.** The variation bound (Theorem 3) does not require this assumption; while it is necessary for small-loss

bound (Theorem 5), because the self-bounding property only holds for non-negative smooth functions.

**Question 2.** "how would the algorithms / results change if $\mathbf{0}$ is not in $\mathcal{X}$?"

**Answer 2.** There are three places we use this assumption:

- Line 544: we use $\|\mathbf{x}_t\|_2 \leq D$ to obtain path-length $P_T$. Note that actually we can bound this term without

assuming $\mathbf{0} \in \mathcal{X}$, by an alternative argument (similar to the analysis of term (ii) in Line 450).

- Line 148: we use $\|\mathbf{x}_{t,i}\|_2 \leq D$ to bound adaptive quantity $D_\infty$. We clarify that we can bound the term

without $\mathbf{0} \in \mathcal{X}$, by alternatively setting surrogate loss as $\ell_{t,i} = \langle f_t(\mathbf{x}_t), \mathbf{x}_{t,i} - \mathbf{x}_t \rangle$ and optimism as $m_{t,i} =$

$\langle f_{t-1}(\bar{\mathbf{x}}_t), \mathbf{x}_{t,i} - \mathbf{x}_t \rangle$ (note that the term $\mathbf{x}_t$ in optimism will be canceled out in the weight update equation).

- Line 150: we use $\|\mathbf{x}_{t,i}\|_2 \leq D$ to bound the term $\|\mathbf{x}_t - \bar{\mathbf{x}}_t\|_2$. It seems necessary to assume $\mathbf{0} \in \mathcal{X}$, and we

do not figure out how to eliminate the requirement yet.

About the self-bounding property of smooth functions, you are absolutely correct. We will resolve it according to your

suggestion. For other minor issues, we will carefully revise the paper according to your constructive comments.

---

**[To Reviewer #4]** Thanks for your review. Below we address your concerns and clarify the misunderstandings.

**Question 1.** "In my opinion, such a paper must demon-

strate the results on several tasks (even on toy problems)."

**Answer 1.** We performed the empirical studies in the re-

buttal period. Due to page limits, we present comparisons

of OGD [1], Ader [10] and Sword (our method) only on

synthetic non-stationary data in Figure 1, which clearly

shows the effectiveness of our algorithms. More results and

elaborations will be included in the revised version.

Figure 1: cumulative loss (left); instantaneous loss (right).

**Question 2.** "The novelty of the paper is limited. The algorithm is a combination of known and well-studied ideas. "

**Answer 2.** We cannot agree with this claim. We highlight the challenge and technical contributions in Line 62–78 of the

paper. Note that a trivial combination of existing meta-algorithms and expert-algorithms cannot obtain desired variation

bounds. We resolve the challenge by carefully designing the optimism for OptimisticHedge used as the meta-algorithm

(see more discussions in Line 122–127 & 146–151). The lower bound for universal dynamic regret is also novel.

We hope the reviewer could check the comments from other reviewers. For example, Reviewer #1 said "The paper

motivates the problem well. The lower bound for regret at least P is nice to have.". Reviewer #2 said "On a technical

level, the authors provide an interesting construction of a new online learning algorithm that brings together key ideas

from various existing procedures.". Reviewer # 3 said "The results will be interesting to the online learning community,

especially given the new interest in dynamic regret and meta-gradient algorithms."

Overall, we believe our work has sufficient novelty and contributions to the online learning community.

[Meta-Review · NeurIPS 2020]

Four knowledgeable referees evaluated the paper. Three recommend acceptance with good arguments. A fourth recommends rejection, but with a very short review claiming in particular that the result can be assembled from existing ingredients. The referee did not respond to either the author rebuttal or my request in the subsequent discussion asking for details. It is my opinion that such a review should at least be substantiated with references, so I am choosing to discount this review.